# Metabolome and Transcriptome Combinatory Profiling Reveals Fluconazole Resistance Mechanisms of *Trichosporon asahii* and the Role of Farnesol in Fluconazole Tolerance

**DOI:** 10.3390/microorganisms11112798

**Published:** 2023-11-17

**Authors:** Xiaoping Ma, Wanling Yang, Aining Yang, Dong Chen, Chengdong Wang, Shanshan Ling, Sanjie Cao, Zhicai Zuo, Ya Wang, Zhijun Zhong, Guangneng Peng, Ming He, Yu Gu

**Affiliations:** 1Key Laboratory of Animal Disease and Human Health of Sichuan Province, College of Veterinary Medicine, Sichuan Agricultural University, Chengdu 611130, China; 2020203075@stu.sicau.edu.cn (W.Y.); wageda369@gmail.com (A.Y.); csanjie@sicau.edu.cn (S.C.); zzcjl@126.com (Z.Z.); wangya13570@sicau.edu.cn (Y.W.); zhongzhijun488@126.com (Z.Z.); pgn.sicau@163.com (G.P.); hemingxx619@163.com (M.H.); 2Sichuan Provincial Center for Animal Disease Prevention and Control, Chengdu 610041, China; cd30408@163.com; 3China Conservation and Research Center for the Giant Panda, Chengdu 611800, China; wangchengdong@aliyun.com (C.W.); lingshanshan86@163.com (S.L.); 4College of Life Sciences, Sichuan Agricultural University, Chengdu 611130, China

**Keywords:** fluconazole resistance, *Trichosporon asahii*, farnesol, metabolome, transcriptome

## Abstract

*Trichosporon asahii* is a basidiomycete yeast that is pathogenic to humans and animals, and fluconazole-resistant strains have recently increased. Farnesol secreted by fungi is a factor that causes variations in fluconazole resistance; however, few studies have explored the underlying mechanisms. Therefore, this study aims to delineate the fluconazole resistance mechanisms of *T. asahii* and explore farnesol’s effects on these processes. A comparative metabolome–transcriptome analysis of untreated fluconazole-sensitive (YAN), fluconazole-resistant (PB) *T. asahii* strains, and 25 μM farnesol-treated strains (YAN-25 and PB-25, respectively) was performed. The membrane lipid-related genes and metabolites were upregulated in the PB vs. YAN and PB-25 vs. PB comparisons. Farnesol demonstrated strain-dependent mechanisms underlying fluconazole tolerance between the YAN and PB strains, and upregulated and downregulated efflux pumps in PB-25 and YAN-25 strains, respectively. Membrane lipid-related metabolites were highly correlated with transporter-coding genes. Fluconazole resistance in *T. asahii* was induced by membrane lipid bio-synthesis activation. Farnesol inhibited fluconazole resistance in the sensitive strain, but enhanced resistance in the resistant strain by upregulating efflux pump genes and membrane lipids. This study offers valuable insights into the mechanisms underlying fungal drug resistance and provides guidance for future research aimed at developing more potent antifungal drugs for clinical use.

## 1. Introduction

*Trichosporon asahii*, a basidiomycete yeast ubiquitously found in nature, is an opportunistic pathogenic fungus that causes trichosporonosis [1,2]. In recent decades, human clinical cases of *T. asahii* infection have been widely reported, and the symptoms include fungemia, granulomas in the lung, chronic meningoventriculitis, intraventricular fungal balls, and superficial infections [3,4,5]. *T. asahii* also causes diseases in animals, such as disseminated fungal infection and fungemia in *Basiliscus plumifrons* [6]. As an important zoonotic fungus, *T. asahii* is commonly encountered in clinical environments, where it can cause trichosporonosis. Therefore, studying this fungus has important research implications.

Fluconazole is one of the main broad spectrum antifungal agents used against *T. asahii* infections. Ergosterol is a major component of the fungal cell membrane [7], and fluconazole’s target site is *erg11*, which is involved in ergosterol biosynthesis [8,9]. However, long-term treatment and repeated administration increase the resistance of *T. asahii* to fluconazole [10,11,12]. The most common mechanisms of fungal multidrug resistance include target site mutations, efflux pump overexpression, and other compensatory stress responses caused by antifungal drugs [13]. In addition, the inhibition of the sterol pathway by fluconazole leads to an increased secretion of farnesol, a sesquiterpene alcohol with an impact on drug resistance [14].

Farnesol is the first quorum sensing molecule described in eukaryotes [15]. The accumulation of farnesol can inhibit the hyphal formation and switch the fungus from the mycelial to the yeast phase [15]. Farnesol can reportedly induce several ultrastructural alterations [16]. Regarding the role of farnesol in antifungal drug resistance, some studies suggest that farnesol promotes antifungal tolerance, whereas others indicate adverse effects [17,18]. Resistance to fluconazole caused by farnesol is a challenge for the treatment of fungal infections; moreover, the exact mechanism involved in this regulation is unclear.

Transcriptome analyses of *T. asahii* have been performed to characterize differentially expressed genes (DEGs) associated with ergosterol biosynthesis and drug transporters in fluconazole-sensitive and fluconazole-resistant strains [19]. Farnesol was found to influence the transcriptome of *Candida auris*, including the upregulation of genes related to antioxidative defense and transmembrane transport [20]. Metabolomic studies have revealed that farnesol has a major influence on the metabolism of *C. albicans* [21]. However, metabolomic analyses of fluconazole-sensitive strains and fluconazole-resistant strains of *T. asahii* are lacking. Thus, combinatorial transcriptomic and metabolomic analysis can contribute to deepening our understanding of the mechanisms underlying *T. asahii* resistance to fluconazole.

In this study, the tolerance mechanisms to fluconazole and the impact of farnesol on drug insensitivity in *T. asahii* were examined by a combination of metabolomics and transcriptomics analyses. Our study can provide novel insights into delineating the mechanisms of fungal drug resistance and can act as a reference basis for the research and development of more efficient antifungal agents.

## 2. Materials and Methods

### 2.1. Strains and Growth Conditions

The *T. asahii* strain that is highly sensitive to fluconazole (YAN) (Genbank accession numbers MN368077.1) was isolated from the vagina of giant pandas (*Ailuropoda melanoleuca*). The YAN strains were cultured on a Sabouraud dextrose agar (SDA) that contained fluconazole with stepwise increasing concentrations from 1 to 16 μg/mL to induce highly resistant strains. The resistant strains were then cultured on a fluconazole-free SDA medium and sequentially passaged. The resultant stable resistant strain after ten subsequent generations was named PB. Farnesol was dissolved in methanol and diluted in yeast extract peptone dextrose medium to 25 µM (the concentration of farnesol was chosen based on pre-experiments). To systematically investigate the potential genes in *T. asahii* that present differences in expression in response to farnesol treatment, 25 µM of farnesol was administered to YAN and PB. After this farnesol treatment, the resulting strains were named YAN-25 and PB-25, respectively. Each sample was cultured at 115× *g* and 25 °C for 48 h until reaching a density of approximately 10^3^ CFU/mL.

### 2.2. Antifungal Susceptibility Testing

Minimum inhibitory concentration (MIC) assays were performed by the broth microdilution (BMD) method, as described in the CLSI document M27-A3 [22]. Suspensions of YAN, PB, YAN-25, and PB-25 were diluted in a Sabouraud dextrose broth and adjusted to the same concentration (1 × 10^3^ CFU/mL). Aliquots (100 μL) of the diluted suspension were inoculated into 96-well microtiter plates containing two-fold serial dilutions of fluconazole. The final concentrations of fluconazole ranged from 0.5 to 256 μg/mL. The MIC endpoints were read at 50% inhibition for fluconazole after 48 h of incubation at 25 °C. The susceptibility tests were determined in triplicate.

### 2.3. Transcriptomic Analysis

An RNA extraction kit was used to extract total RNA (Tiangen Biotech Co., Ltd., Beijing, China). RNA integrity and DNA contamination were analyzed via agarose gel electrophoresis. RNA purity was measured using a NanoDrop spectrophotometer (Thermo Scientific, Wilmington, NC, USA); RNA concentrations were quantified using Qubit (Thermo Scientific, Wilmington, NC, USA), while RNA integrity was precisely measured in an Agilent 2100 Bioanalyzer system (Agilent Technologies, Santa Clara, CA, USA) [23,24]. The Illumina HiSeq sequencing platform was used to (a) build a database for sequencing, (b) filter the original sequence data to obtain clean reads, (c) count the quality indicators of clean reads, (d) evaluate the sequencing quality, and (e) compare the data to the reference genome [25]. The reads were processed and filtered using the SMRT Link version 4.0 software [26]. The Q20 and Q30 percentages for each sample were greater than 97% and 94%, respectively. The raw reads were mapped to the reference genome using the GMAP v. 2020-6-30. RSEM v. 1.3.0 was used to calculate the number of each gene [27]. Specified pairwise transcriptome comparisons (PB vs. YAN; PB-25 vs. YAN-25; YAN-25 vs. YAN; and PB-25 vs. PB) were performed to identify the main DEGs with |log_2_ FC| ≥ 1 and *p*-value < 0.05. A Venn diagram was used to show the comparison and overlap between DEGs in the three comparisons. KEGG pathway and GO analyses were conducted using the R package “clusterProfiler” (version 3.18) for DEGs [28]. Each group contained three independent biological replicates.

### 2.4. Metabolomic Analysis

Culture samples were centrifuged at 12,480× *g* at 4 °C for 10 min, and 100 µL of supernatant was mixed with 800 µL of a methanol–water solution (1/: 4, vol/: vol) for 15 min before lyophilization; finally, the dried samples were stored at −80 °C for subsequent LC-MS assays [29]. The column temperature was 45 °C with a flow rate of 0.4 mL/min; the injection volume was 1 µL; and the mobile phases were water (A) and acetonitrile/methanol 2:3 (*v/v*) (B) containing 0.1% (*v/v*) of formic acid. The linear gradient was as follows: 0 min, 1% B; 1 min, 30% B; 2.5 min, 60% B; 6.5 min, 90% B; 8.5 min, 100% B; 10.7 min, 100% B; 10.8 min, 1% B; and 13 min, 1% B. Raw data handling was performed using the Progenesis QI software v.3.0 (Waters Corporation, Milford, CT, USA). Metabolites with variable importance plot values larger than 1.0 and *p*-value less than 0.05 were considered differential metabolites. The metabolomics analyses were repeated six times.

### 2.5. Integrative Analysis of Metabolomic and Transcriptomic Data

The DEMs related to membrane lipids and DEGs implicated in efflux pump and membrane lipids were selected for integrative analysis. Correlation analysis was performed using the OmicStudio tools at https://www.omicstudio.cn/tool/62 (accessed on 9 July 2023). DEMs and DEGs were considered significantly correlated when |r| > 0.8 and *p*-values < 0.5. A correlation network was generated using the OmicStudio tools at https://www.omicstudio.cn/tool (accessed on 9 July 2023).

## 3. Results

### 3.1. Effect of Farnesol on Fluconazole Resistance in T. asahii

In order to assess the effects of farnesol concentration on fluconazole resistance in *T. asahii*, YAN and PB were treated by different concentrations of farnesol and the MIC was determined using a broth dilution. The results showed that 25 μM of farnesol has the greatest impact on the fluconazole resistance of YAN and PB. Therefore, we chose 25 μM of farnesol for further research (Appendix A). The MIC of fluconazole showed that although farnesol had little effect on the sensitivity of the YAN strain to fluconazole, it further increased the fluconazole tolerance of the PB strain. This trend indicated that farnesol might have different mechanisms of action in YAN and PB. To further test this hypothesis, transcriptomic and metabolomic analyses of the YAN, PB, YAN-25, and PB-25 strains were performed (Table 1).

### 3.2. Analysis of RNA-Seq Data

In this study, we performed RNA-seq analyses of 12 samples (three biological replicates for each group) and obtained 721,405,882 bp of raw reads and 692,727,350 bp of clean reads. The error rate was 0.02%, suggesting that the sequence data were accurate. The GC content ranged from 61.58 to 62.29% (Appendix A). These data demonstrate that the transcriptome dataset was reliable. Among the 12 samples, 89.21–91.49% of the clean reads were mapped to the reference genome, and 88.33–90.98% were uniquely mapped (Appendix A).

### 3.3. Differential Expression of Genes between Four Pairwise Strain Combinations

Transcriptomic comparisons were performed to identify DEGs. The specific pairwise comparisons were as follows: (1) PB vs. YAN, (2) YAN-25 vs. YAN, (3) PB-25 vs. PB, and (4) PB-25 vs. YAN-25. There were 1703 DEGs in PB vs. YAN group (570 upregulated and 1132 downregulated) and 960 DEGs in PB-25 vs. YAN-25 (327 upregulated and 633 downregulated). However, there were only 279 DEGs in the YAN-25 vs. YAN (159 upregulated and 120 downregulated) and 699 DEGs in the PB-25 vs. PB (602 upregulated and 97 downregulated) groups (Appendix A). These results indicate that the strain with induced resistance generated more DEGs relative to the level of farnesol use. The effect of farnesol on resistant strains was greater than that on wild-type strains. Most of the DEGs were downregulated following the development of fluconazole resistance. In contrast, there were more upregulated genes in YAN and PB following farnesol treatment. Venn diagram analysis showed that there were 12 DEGs that were common to the four comparisons (Appendix A).

### 3.4. Gene Ontology (GO) Enrichment and Pathway Analysis of Transcriptomic Data

The GO analysis investigated the roles and enrichment of the DEGs (Appendix A). In the biological process (BP) categories, the DEGs identified in the four groups were all enriched in “transport (GO:0006810)”, “transmembrane transport (GO:0055085)”, “localization (GO:0051179)”, and “establishment of localization (GO:0051234)”. In the cellular component (CC) category, the DEGs of the four comparisons were primarily enriched in “intrinsic component of membrane (GO:0031224)” and “membrane part (GO:0044425)”. In the molecular function (MF) category, the DEGs in the four comparisons were enriched in “transmembrane transporter activity (GO:0022857)”. Briefly, several GO terms associated with the membrane were enriched across all four comparisons. Therefore, DEGs associated with membrane component localization and transmembrane transport might play important roles in fluconazole resistance and farnesol action in *T. asahii*.

Furthermore, to evaluate the functions and biological pathways represented by the DEGs, a Kyoto Encyclopedia of Genes and Genomes (KEGG) functional analysis was performed. As shown in Figure 1a, the DEGs of PB vs. YAN were involved in ribosome, tryptophan metabolism, and pentose and glucuronate interconversions. For YAN-25 vs. YAN, DEGs were assigned to oxidative phosphorylation and propanoate metabolism (Figure 1b). However, the DEGs of PB-25 vs. PB were involved in autophagy—yeast and pyruvate metabolism (Figure 1c). For the PB-25 vs. YAN-25 combination, the enriched pathways included butanoate, starch, and sucrose metabolism (Figure 1d).

Notably, although there were distinct differences in the top 20 enriched pathways across the four comparisons, several lipid-related pathways, such as glycerolipid metabolism, were only enriched in YAN-25 vs. YAN and PB-25 vs. PB. It was inferred that farnesol treatment might have some effects on lipid-related pathways in *T. asahii*.

### 3.5. Identification of Genes Associated with Efflux Pumps

Based on the GO analysis results, the research focused on DEGs involved in “transmembrane transport” and “transport”. Most of the DEGs belonging to these GO terms were members of the major facilitator superfamily (MFS) family. The expression level of the genes related to the ATP-binding cassette (ABC) and MFS transporters were represented in heatmaps (Figure 2). The results showed that the genes associated with the efflux pump were predominantly downregulated in PB vs. YAN and PB-25 vs. YAN-25, implying that decreasing efflux capacity is an important factor in the development of drug resistance to *T. asahii.* However, most of the transporter-coding genes were upregulated following treatment with farnesol in PB-25 vs. PB but downregulated in the YAN-25 vs. YAN group. This finding is consistent with the MIC results and implies that farnesol has different mechanisms of action for drug-sensitive and drug-resistant *T. asahii* strains, and can increase resistance of the drug-resistant strains.

### 3.6. Identification of the Genes Associated with Membrane Lipids

Based on the KEGG enrichment analysis, the DEGs associated with membrane lipids were identified and visualized using heatmaps (Figure 3). The results for both the PB vs. YAN and PB-25 vs. PB comparisons showed that the genes associated with membrane lipids were predominantly upregulated. However, the other two comparisons did not show a similar trend. The changes in the expression level of these genes indicated that a continuous induction of fluconazole altered the expression of the genes related to membrane lipid, thereby leading to drug resistance in PB. Farnesol treatment upregulated membrane lipid-related genes in PB but did not have the same impact on YAN. Therefore, the increase in fluconazole resistance after exposure to farnesol only occurred in PB and not in YAN.

### 3.7. Differentially Accumulated Metabolites among the Four Comparisons

To investigate the changes in metabolites, differential metabolite analyses were performed for the comparisons of PB vs. YAN, YAN-25 vs. YAN, PB-25 vs. PB, and PB-25 vs. YAN-25 groups. As shown in Appendix A, we identified a total of 314 significant differentially expressed metabolites (DEMs) in PB vs. YAN (218 upregulated and 96 downregulated), 128 in PB-25 vs. YAN-25 (35 upregulated and 93 downregulated), 193 in YAN-25 vs. YAN (91 upregulated and 102 downregulated), and 283 in PB-25 vs. PB (104 upregulated and 179 downregulated). The changes in metabolites were consistent with the changes in gene expression in the transcriptome. The Venn diagram showed only one DEM shared among the four comparisons (Appendix A).

### 3.8. Pathway Enrichment of DEMs

The KEGG pathways were used to investigate the DEM functions. The DEMs of PB vs. YAN were significantly involved in linoleic acid metabolism, purine metabolism, ABC transporters, and amino acid biosynthesis. For YAN-25 vs. YAN, DEMs were assigned to linoleic acid metabolism, lysine degradation, and arginine biosynthesis. For PB-25 vs. PB, the enriched pathways included linoleic acid metabolism, amino acid biosynthesis, TCA cycle, and lysine degradation. Finally, the DEMs of PB-25 vs. YAN-25 were involved in purine metabolism, linoleic acid metabolism, and monoterpenoid biosynthesis (Figure 4). Although the KEGG pathway enrichment varied distinctly among the four combinations, linoleic acid metabolism was the most enriched pathway. Farnesol led to an enrichment of DEMs in PB strain pathways, suggesting that it had a greater effect on the resistant strains. The pathways affected by farnesol in terms of fatty acid metabolism included “linoleic acid metabolism”, “arachidonic acid metabolism”, “alpha-linolenic acid metabolism”, “fatty acid biosynthesis”, “biosynthesis of unsaturated fatty acids”, and “lipoic acid metabolism”.

### 3.9. DEMs Related to Membrane Lipids

In this study, many metabolites belonging to the classes of glycerolipids, glycerophospholipids, and sterol lipids were distinctly upregulated following tolerance development to fluconazole in PB vs. YAN (Figure 5a). Several phosphatidylserines were significantly downregulated in PB-25 vs. PB, whereas glycerophosphoethanolamines, glycerolipids, and sterol lipids showed the opposite trend (Figure 5b). In the YAN-25 vs. YAN and PB-25 vs. YAN-25 comparisons, no marked differences were observed. These observations are consistent with our transcriptome analysis results. The metabolome results further demonstrated that the upregulation of membrane lipid leads to fluconazole resistance in PB. Farnesol treatment could only alter the composition of membrane lipids in PB but not in YAN, which has been a contributing factor in the increase in the MIC value in PB-25 and decrease in YAN-25.

### 3.10. Association Analysis of DEMs and DEGs

To better characterize the relationship between the DEGs and DEMs implicated in membrane lipid and efflux pump regulation, correlation network maps between the metabolomics and transcriptomics were constructed. Notably, in PB vs. YAN, PS (O-20:0/18:4 (6Z,9Z,12Z,15Z)) was correlated with the highest number of DEGs and negatively correlated with 52 efflux pump-related DEGs (Figure 6a). In PB-25 vs. PB, LysoPE(20:2(11Z,14Z)/0:0) was correlated with the highest number of DEGs and positively correlated with five efflux pump-related DEGs (Figure 6b). The above results suggest that efflux transporters might be associated with the membrane lipids in *T. asahii*.

## 4. Discussion

Fungal resistance to azole drugs has recently become a growing concern. As a common opportunistic fungus, *T. asahii* has shown increasing resistance against fluconazole over the last two decades [8,30,31]. Destruction of the cell membrane is the primary mechanism of the inhibition of fungal growth by azole compounds, which could decrease ergosterol synthesis by inhibiting lanosterol 14α-demethylase encoded by *erg11* [32]. In addition, many cytochrome-dependent enzymes are inhibited by azoles [33]. To mediate the effects of the antifungal mechanisms, fungi may enhance resistance to azoles by increasing the expression of genes and metabolites related to membrane lipids and efflux pumps (Appendix A). Farnesol produced by fungi has been proven to inhibit the development of fluconazole resistance in *C. albicans* by partially affecting gene expression during ergosterol biosynthesis [34]. However, other studies have reported that the upregulation of drug efflux pumps mediated by farnesol can lead to enhanced tolerance [17]. To explore the fluconazole resistance mechanism and farnesol’s effects on *T. asahii*, gene expression (metatranscriptomics) and metabolic profiles (metabolomics) should be analyzed. 

In this study, a comparison of fluconazole resistance in YAN, PB, YAN-25, and PB-25 was performed based on an MIC test, and the results showed that farnesol had different effects on fluconazole resistance in the fluconazole-sensitive YAN and fluconazole-resistant PB strains. These results may be at least partially attributed to the different mechanisms of action of farnesol in different strains. Our transcriptomic and metabolomic results further support this hypothesis. As shown in the corresponding Venn diagrams, only 73 DEGs (Appendix A) and 13 DEMs (Appendix A) were shared in the YAN-25 vs. YAN and PB-25 vs. PB groups, and most of these DEGs and DEMs did not overlap. Volcano plots (Appendix A) demonstrated that the number of DEGs in YAN-25 vs. YAN was significantly lower than that in the other three comparisons. This may explain why farnesol could only induce small changes in fluconazole resistance in the YAN strain.

Several reports suggest that increasing the cellular levels of membrane lipids and activation of drug efflux pumps could be part of the defense response of *T. asahii* to fluconazole-induced stress [8,30,31]. Farnesol may prompt a protective stress response in the fungal cell, which can indirectly lead to the development of tolerance to the compound [17]. One of the fundamental mechanisms of action through which farnesol inhibits fungal growth is the attenuation of ergosterol synthesis [35]. Therefore, the upregulation of genes and metabolites related to membrane lipids in the drug-resistant strain might lead to an antagonistic response that negates or reduces the inhibitory effect of farnesol on *T. asahii*. Membrane lipids in fungi mostly include glycerolipids, glycerophospholipids, sphingolipids, and sterols [36]. Metabolome and transcriptome analyses showed that the genes and metabolites related to membrane lipids were markedly upregulated in the PB vs. YAN and PB-25 vs. PB comparisons. The upregulated genes included *erg28* (A1Q1-02746), *mvd* (A1Q1-07715), and *erg11* (A1Q1-02098) (Figure 3). Another essential gene for ergosterol biosynthesis, *erg28,* encodes a scaffold protein in yeast, which connects a group of ergosterol synthetases to form a sterol C4 demethylase (SC4DM) multi-enzyme complex [30]. Our metabolomics results further revealed that sterol lipids, glycerolipids, and glycerophospholipids were upregulated following the mounting of fluconazole tolerance responses in PB vs. YAN and PB-25 vs. PB. Both transcriptomics and metabolomics indicated that farnesol treatment may lead to an increased synthesis of membrane lipids in PB and might be a possible reason for the elevated fluconazole resistance in *T. asahii*.

Drug export by efflux pumps is a common mechanism underlying antimicrobial drug resistance [31]. Efflux pump proteins are classified into two families: the ABC and MFS families [37]. These proteins are classified based on the source covering their energy requirements. The energy source for ABC transporters is ATP and its subsequent hydrolysis [38], whereas that of MFS members is proton motive force [39]. Notably, ABC or MFS transporter-related DEGs were downregulated in the PB vs. YAN and PB-25 vs. YAN-25 comparisons (Figure 2A). Most MFS efflux pump-encoding genes were downregulated in YAN-25 vs. YAN but upregulated in PB-25 vs. PB (Figure 2B,C). Based on the above results, we hypothesized that farnesol could cause a stress reaction that led to the upregulation of drug efflux pumps in PB, which could in turn contribute to the increased fluconazole resistance of the PB-25 strain that we examined in our work. However, the genes related to efflux pump were downregulated following farnesol treatment in YAN, resulting in a lower MIC value in YAN-25 than in YAN. This may at least partially explain the different effects of farnesol regarding fluconazole resistance in the YAN and PB strains in our experiments.

To further observe the associations of the efflux pump and membrane lipid, correlation network diagrams were constructed (|r| > 0.90) to investigate the connection between DEGs and DEMs in PB vs. YAN and PB-25 vs. PB (Figure 6). The correlation analysis suggested that membrane lipid-related metabolites were highly correlated with transporter-coding genes. Studies on drug resistance have demonstrated that membrane lipids could modulate the expression and activity of efflux pumps. Phospholipids and cholesterol in the membrane could promote or inhibit drug efflux by binding to the transmembrane helices of ABC transporters and altering their conformation [40]. In addition to binding to the transmembrane helices, membrane lipids can combine with the substrate chamber of ABC transporters and drive substrates toward the transporter cavity [41]. The effects of membrane lipids on efflux pumps might contribute to the rise in fluconazole resistance in PB strains.

Our study has a few limitations. First, the overall design of our study did not allow us to establish a causal relationship between membrane lipids and efflux pumps in *T. asahii*, nor could we identify any changes in the composition of membrane lipids that may have caused alterations in the transmembrane transporters or vice versa. Second, this study solely focused on studying one specific concentration of farnesol; however, varying concentrations of farnesol might have different effects on fungal fluconazole resistance.

## 5. Conclusions

In this study, a combined transcriptomic and metabolomic analysis was conducted to investigate the effects of farnesol on fluconazole resistance in *T. asahii*. Antimicrobial susceptibility testing results showed that farnesol reduced the MIC of the fluconazole-sensitive strain but markedly increased the innate tolerance of the fluconazole-resistant strain. Farnesol may have different mechanisms of action in fluconazole-sensitive and resistant *T. asahii* strains. The upregulation of membrane lipid metabolism may be strongly associated with fluconazole resistance in the resistant strain compared to that in the sensitive strain. After farnesol treatment, the decrease in fluconazole resistance in the sensitive strain may be due to the inhibition of farnesol on ergosterol synthesis and the downregulation of the efflux pump, while the increase in fluconazole resistance in the resistant strain may be due to the overexpression of the efflux pump and membrane lipids. This study provides valuable insights into aiding research toward delineating the mechanism of antifungal tolerance of *T. asahii*, and the role of farnesol in fluconazole resistance could also contribute to ongoing and future efforts in developing novel and more potent antifungal drugs for use in clinical settings.

## Figures and Tables

**Figure 1 microorganisms-11-02798-f001:**
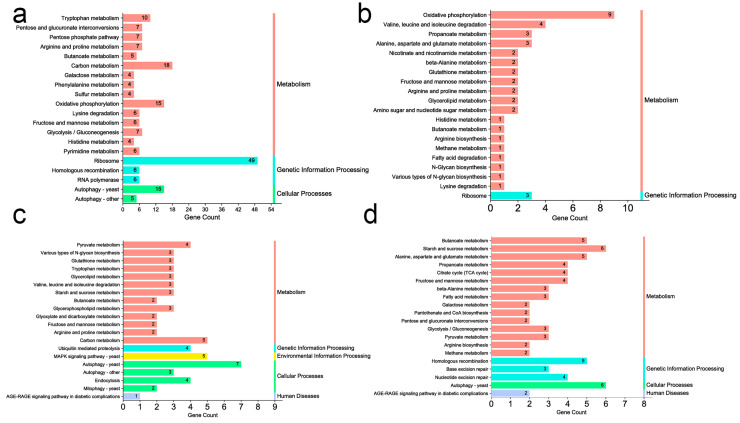
KEGG enrichment analysis of DEGs in (**a**) PB vs. YAN, (**b**) YAN-25 vs. YAN, (**c**) PB-25 vs. PB, and (**d**) PB-25 vs. YAN-25. The Y-axis represents the KEGG pathway, and the X-axis shows the number of DEGs in each pathway.

**Figure 2 microorganisms-11-02798-f002:**
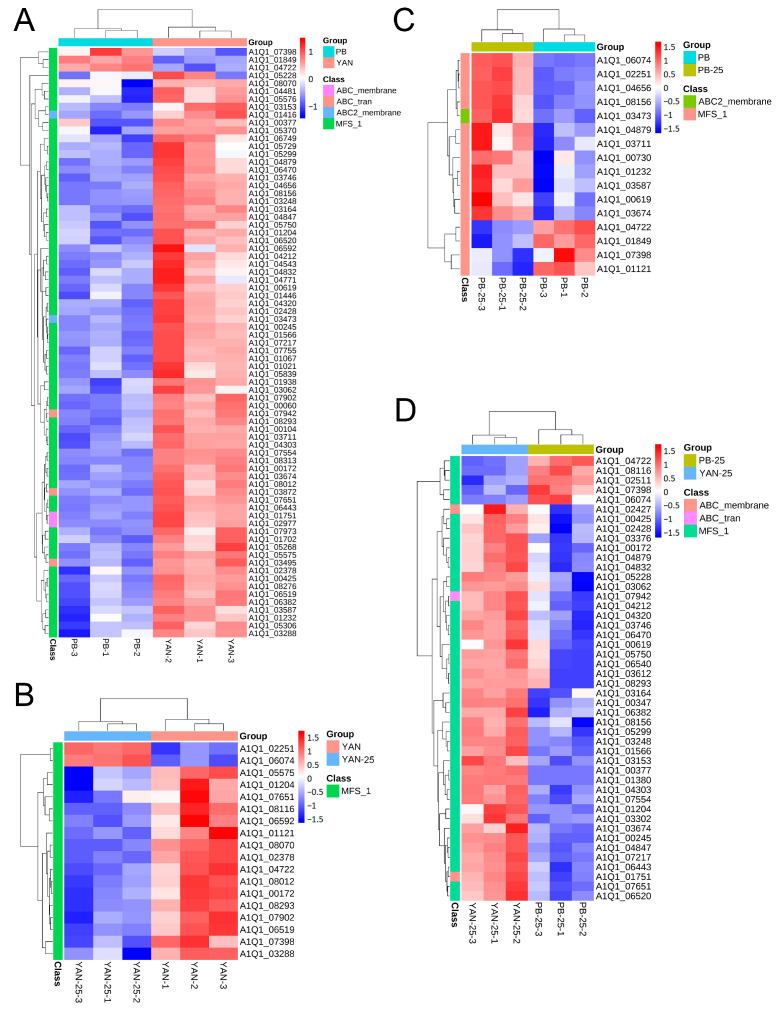
Heatmap of the DEGs implicated in efflux pump regulation among (**A**) PB vs. YAN, (**B**) YAN-25 vs. YAN, (**C**) PB-25 vs. PB, and (**D**) PB-25 vs. YAN-25. Blue indicates low expression levels, and red indicates high expression levels. Colored blocks along the Y-axis of the heatmap represent the family to which each gene belongs.

**Figure 3 microorganisms-11-02798-f003:**
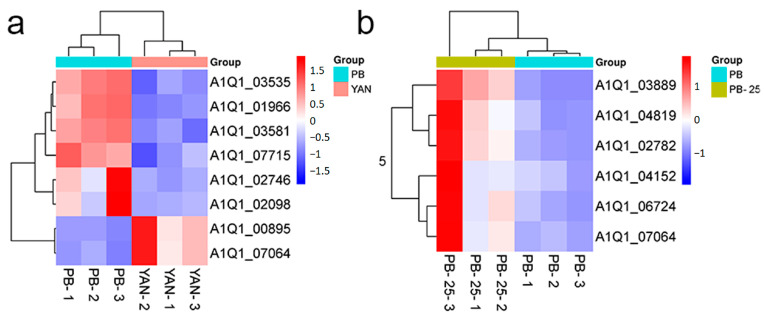
Heatmap of the DEGs implicated in membrane lipids based on the (**a**) PB vs. YAN and (**b**) PB-25 vs. PB comparisons. Blue indicates low expression levels, and red indicates high expression levels.

**Figure 4 microorganisms-11-02798-f004:**
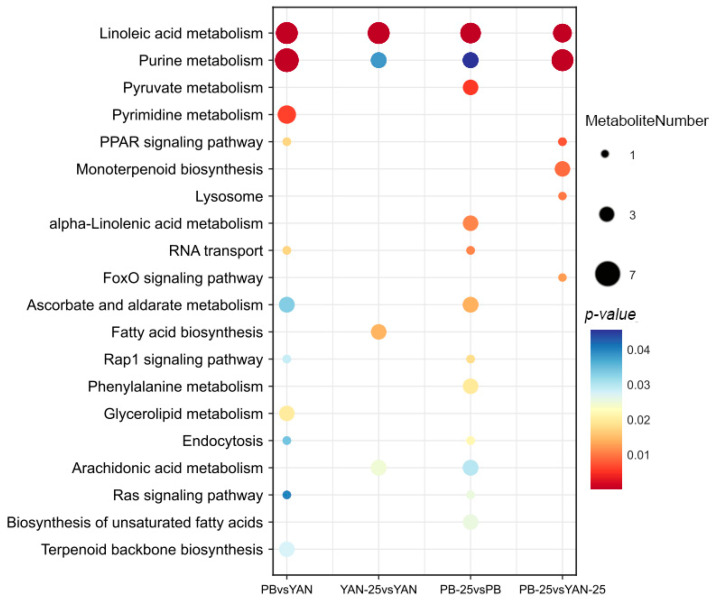
KEGG pathway assignment of differentially expressed metabolites among PB vs. YAN, YAN-25 vs. YAN, PB-25 vs. PB, and PB-25 vs. YAN-25. The dot color represents the *p*-value, and the dot size represents the number of differentially expressed metabolites.

**Figure 5 microorganisms-11-02798-f005:**
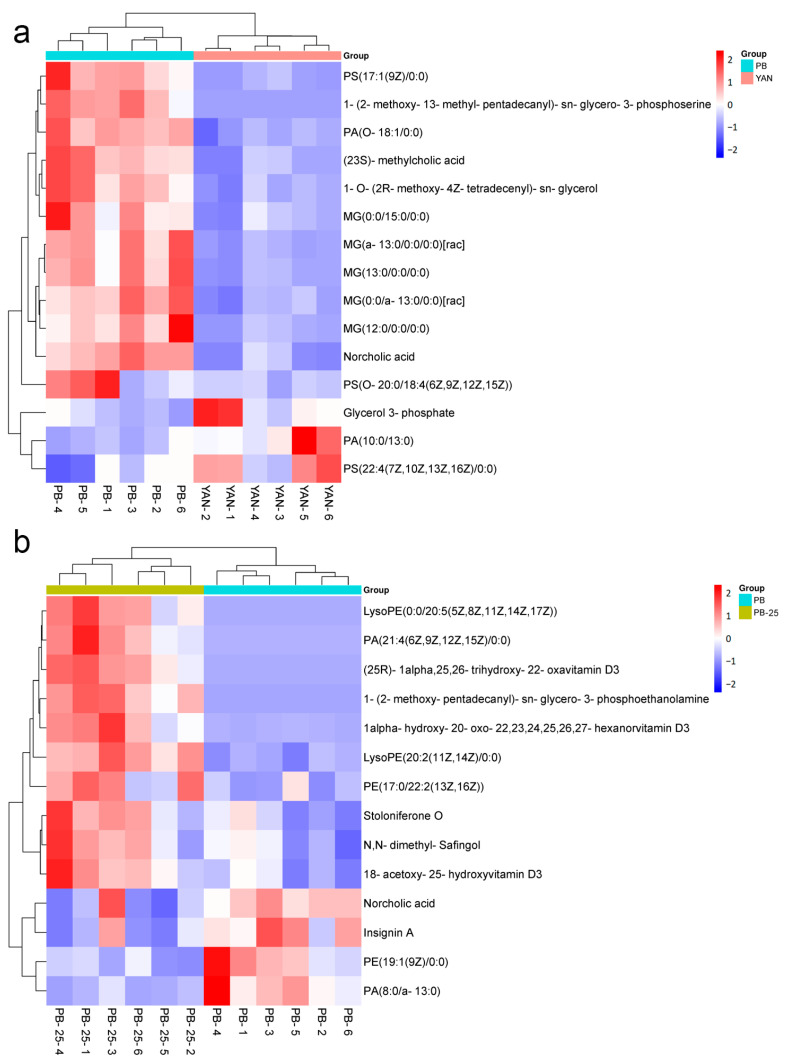
Heatmap of the differentially expressed metabolites implicated in membrane lipids among (**a**) PB vs. YAN and (**b**) PB-25 vs. PB. Blue indicates low expression levels, and red indicates high expression levels.

**Figure 6 microorganisms-11-02798-f006:**
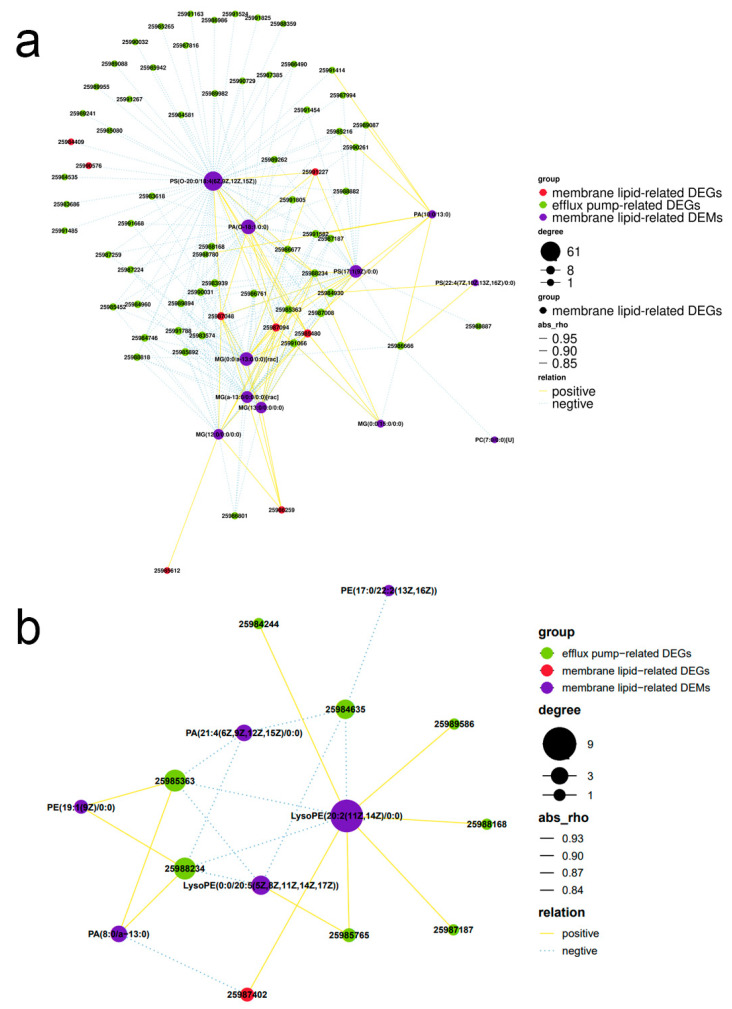
Association analysis of transcriptomic and metabolomic variations. Connection network between screened differentially expressed metabolites and DEGs in (**a**) PB vs. YAN and (**b**) PB-25 vs. PB. Red and purple ovals in the nodes represent differentially expressed metabolites and DEGs, respectively. The size of each node indicates the degree of betweenness centrality. The lines represent the “relationships” between differentially expressed metabolites and DEGs; yellow and blue represent positive and negative correlations, respectively; and the thickness of the lines indicates the strength of the relationship.

**Table 1 microorganisms-11-02798-t001:** In vitro susceptibilities of YAN, YAN-25, PB, and PB-25 to fluconazole.

Strain	MIC (μg/mL)
YAN	2
YAN-25	1
PB	32
PB-25	128

## Data Availability

All raw data for RNA-seq were deposited into NCBI (BioProject accession code PRJNA941075).

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
