# Peer review of "Metabolome and Transcriptome Combinatory Profiling Reveals Fluconazole Resistance Mechanisms of Trichosporon asahii and the Role of Farnesol in Fluconazole Tolerance"

_microorganisms, 2023, doi:10.3390/microorganisms11112798_

Round 1

Reviewer 1 Report

Comments and Suggestions for Authors

In this article and Ma et al. (2023), the authors investigated the metabolic and transcriptional profile of Trichosporon asahii resistance to Fluconazole and its relationship with Farnesol. Some points to be answered before publishing this article.

1) line 22: are these the codes for the comparisons?

2) line 78: What is the code for this strain? Or the original work that isolated it?

3) line 188: there seems to be unnecessary information about spinning?

4) lines 80-81: what is the initial and final concentration used?

Author Response

Martin von Bergen                                                        NOV 11st, 2023                                                                                              

Editor 

Microorganisms

Dear Martin von Bergen,

Thank you for giving us the opportunity to revise our manuscript (Manuscript ID: microorganisms-2673574) entitled “Metabolome and Transcriptome Combinatory Profiling Reveals Fluconazole Resistance Mechanisms of Trichosporon asahii and the Role of Farnesol in Fluconazole Tolerance”. The manuscript has been revised according to your suggestions. Our point-to-point responses are attached at the end of this letter.

We hope this revision is satisfactory and we look forward to hearing from you soon. 

With kind regards,

Yu Gu and Xiaoping Ma

Response to Reviewer 1 Comments

1. Summary

2. Questions for General Evaluation

Reviewer’s Evaluation

Response and Revisions

Does the introduction provide sufficient background and include all relevant references?

Yes

Are all the cited references relevant to the research?

Yes

Is the research design appropriate?

Yes

Are the methods adequately described?

Can be improved

Are the results clearly presented?

Yes

Are the conclusions supported by the results?

Yes

3. Point-by-point response to Comments and Suggestions for Authors

Comments 1: line 22: are these the codes for the comparisons?

Response 1: Yes. There are four comparisons in this study: PB vs. YAN, YAN-25 vs. YAN, PB-25 vs. PB, and PB-25 vs. YAN-25. The membrane lipid-related genes and metabolites were upregulated in the PB vs. YAN and PB-25 vs. PB comparisons.

Comments 2: line 78: What is the code for this strain? Or the original work that isolated it?

Response 2: Thank you for pointing this out. The code for this strain is MN368077.1. You can search for detailed information about this strain on this website: https://www.ncbi.nlm.nih.gov/nuccore/MN368077.1/. We have added the code for this strain in line 78-79.

Comments 3: line 188: there seems to be unnecessary information about spinning?

Response 3: We agree with this comment. Therefore, we have deleted the unnecessary information in line 188.

Comments 4: lines 80-81: what is the initial and final concentration used?

Response 4: Thank you for pointing this out. The sensitive strain YAN0802 was induced into resistance strain by 1/2 minimal inhibitory concentration (MIC) induced trails of fluconazole. So the initial and final concentration of fluconazole in SDA were 1 and 16 μg/mL, respectively. We have added information about the initial and final concentration in line 81.

Reviewer 2 Report

Comments and Suggestions for Authors

Review:

The study aimed to delineate the fluconazole resistance mechanisms of the pathogenic basidiomycete yeast T. asahii and explore the effect of farnesol, secreted by fungi as a factor that causes variations in fluconazole resistance, on these resistance processes.

Major comments:

Why did you choose 25mM of farnesol, specifically? The logic behind this choice needs to be addressed in some way, or else it appears arbitrary.

25 .M farnesol-treated strains (YAN-25 and PB-25, respectively) was performed

3.2. Analysis of RNA-Seq Data – of the twelve samples, how many were technical/biological repeats? Please specify it clearly.

Analysis of RNA-Seq Data the data regarding Q20/Q30 from this section can be moved to methods.

Figure 1 should be moved to supplementary data – the more interesting analysis is the enrichment of the KEGG analysis, just GO analysis, though a "good to have" is less informative.

Figure 8 – though lovely, this figure too can be moved to supplementary, as there are already many figures in this paper.

Minor comments:

Table 1 – this is strictly aesthetic issue – I would prefer the table to be elongated and not horizontal. It is just a little harder to read.

Figure 3- It would be best, if possible, that the colors remain the same in all heatmaps. For example, in figure 3a the PB is green, in 3b it is indicated by light blue, this is confusing. The other figures maintain the same color scheme for the same types and it is easier to follow.

Author Response

Martin von Bergen                                                        NOV 12st, 2023                                                                                              

Editor 

Microorganisms

Dear Martin von Bergen,

Thank you for giving us the opportunity to revise our manuscript (Manuscript ID: microorganisms-2673574) entitled “Metabolome and Transcriptome Combinatory Profiling Reveals Fluconazole Resistance Mechanisms of Trichosporon asahii and the Role of Farnesol in Fluconazole Tolerance”. The manuscript has been revised according to your suggestions. Our point-to-point responses are attached at the end of this letter.

We hope this revision is satisfactory and we look forward to hearing from you soon. 

With kind regards,

Yu Gu and Xiaoping Ma

Response to Reviewer 2 Comments

1. Summary

2. Questions for General Evaluation

Reviewer’s Evaluation

Response and Revisions

Does the introduction provide sufficient background and include all relevant references?

Yes

Are all the cited references relevant to the research?

Yes

Is the research design appropriate?

Can be improved

Are the methods adequately described?

Can be improved

Are the results clearly presented?

Can be improved

Are the conclusions supported by the results?

Can be improved

3. Point-by-point response to Comments and Suggestions for Authors

Comments 1: Why did you choose 25μM of farnesol, specifically? The logic behind this choice needs to be addressed in some way, or else it appears arbitrary.

Response 1: Thank you for pointing this out. We have compared the effects of different concentrations of farnesol (0 to 50 μM) on the fluconazole resistance in T. asahii in previous studies. The results showed that 25 μM of farnesol has greatest impact on the fluconazole resistance of YAN and PB. Therefore, we chose 25 μM of farnesol for further research. We have added information about the pre-experiment in line 85-86 and line 141-145.

Comments 2: 3.2. Analysis of RNA-Seq Data – of the twelve samples, how many were technical/biological repeats? Please specify it clearly.

Response 2: Each group contained three independent biological replicates. The number of repetitions have been indicated in line 156-157.

Comments 3: Analysis of RNA-Seq Data – the data regarding Q20/Q30 from this section can be moved to methods.

Response 3: We agree with this comment. Therefore, we have moved the data regarding Q20/Q30 to line 111-112.

Comments 4: Figure 1 should be moved to supplementary data – the more interesting analysis is the enrichment of the KEGG analysis, just GO analysis, though a "good to have" is less informative.

Response 4: Thank you for pointing this out. We have moved this figure to supplementary data and changed its number to Figure S3.

Comments 5: Figure 8 – though lovely, this figure too can be moved to supplementary, as there are already many figures in this paper.

Response 5: We agree with this comment. Therefore, we have moved Figure 8 to supplememtary. The number of this figure has been changed to Figure S5.

Comments 6: Table 1 – this is strictly aesthetic issue – I would prefer the table to be elongated and not horizontal. It is just a little harder to read.

Response 6: Thank you for pointing this out. I have adjusted the table to be elongated in line 153.

Comments 7: Figure 3- It would be best, if possible, that the colors remain the same in all heatmaps. For example, in figure 3a the PB is green, in 3b it is indicated by light blue, this is confusing. The other figures maintain the same color scheme for the same types and it is easier to follow.

Response 7: We agree with this comment. Therefore, we have adjusted the color scheme in Figure 2 (line 222), Figure 3 (line 238), and Figure 5 (line 284).
